# FIKSURVEY: AN AUTOMATED PEER REVIEW LOOP TO RAISE THE CEILING ON AI ACADEMIC WRITING

## ABSTRACT

The escalating demand for comprehensive literature surveys in rapidly evolving research areas makes manual writing increasingly impractical, underscoring the necessity of automation. Large Language Models (LLMs) provide a promising foundation for this task, yet guiding them to generate accurate, reliable content remains a fundamental challenge, as issues such as hallucinations and vague organization often persist. To address this, we propose **FIKSurvey**, a feedback-driven framework grounded in the idea that "*Feedback is the key for automatic survey generation*." Specifically, FIKSurvey systematically incorporates feedback across three dimensions: outline feedback for structural clarity, citation feedback for evidence validation, and content feedback for readability and analytical depth. The framework also supports optional human-in-the-loop intervention for user-specific needs. Experiments confirm that FIKSurvey substantially improves both citation recall and content quality, demonstrating feedback as the critical mechanism for automatic survey generation.

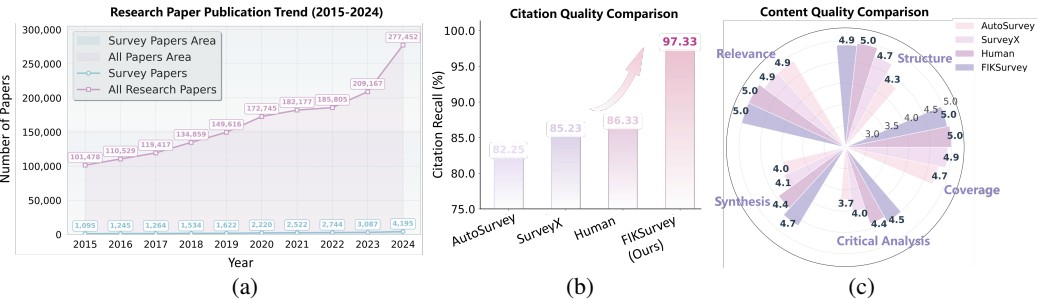

Figure 1: Overview of FIKSurvey. (a) illustrates our motivation to automate the survey generation. The exponential growth of scientific publications underscores the urgent need for automation in survey writing. (b) and (c) indicate the performance improvements achieved by our FiKSurvey.

## 1 INTRODUCTION

The exponential growth of scholarly literature has created a pressing need for automated tools for knowledge synthesis (Bornmann et al., 2021; Fire & Guestrin, 2019), with Large Language Models (LLMs) emerging as a powerful technology for this task (Brown et al., 2020; Raffel et al., 2020; Wei et al., 2022). Recent systems such as AutoSurvey (Wang et al., 2024d) and SurveyX (Liang et al., 2025) represent pioneering efforts in this direction (Lála et al., 2023; Wang et al., 2024b;a). AutoSurvey introduced a multi-stage workflow, while SurveyX advanced the paradigm by incorporating techniques such as AttributeTree representation. These works mark substantial progress, yet their overall paradigm remains linear: information is passed forward through a pipeline, with only shallow or post-hoc refinement (Wei et al., 2022; Kojima et al., 2022).

This linear paradigm contrasts with the core mechanism that ensures quality in human academic writing: the iterative, critical loop of peer review (Bender et al., 2021; Tennant & Ross-Hellauer, 2020). It treats writing as a unidirectional assembly line rather than a dynamic optimization process (Wei et al., 2022; Yao et al., 2023b). Lacking the capacity for self-scrutiny and correction,

surveys generated under this paradigm entrench the flaws from each stage, culminating in weak logical structures, superficial content analysis, and, most critically, factual hallucinations (Ji et al., 2023; Zhang et al., 2019; Manakul et al., 2023; Liang et al., 2022).

Inspired by the rigor of human academic peer review, this paper proposes a paradigm shift from linear generation to a closed-loop, feedback-driven process. We argue that to elevate LLMs from mere text generators to reliable academic authors, the core principles of peer review must be internalized into their operational core. To this end, we propose **FIKSurvey** based on the idea that "*Feedback is the key for automatic survey generation*". This framework automates the peer review cycle through a dual-agent system: a Writer LLM responsible for content creation and a Helper LLM that acts as an automated **"peer reviewer"**. This "reviewer" provides continuous, structured feedback across three meticulously designed dimensions that mirror the core concerns of human reviewers:

1. **Outline Feedback** (Structural Integrity): Just as a human reviewer first assesses a paper's logical skeleton, our framework begins by refining the survey's outline. This ensures a clear, coherent, and academically sound structure *before* content is generated, preventing fundamental logical flaws at the outset.

2. **Content Feedback** (Analytical Depth): Moving beyond simple summarization, this feedback loop pushes the "Writer LLM" to achieve deeper synthesis and critical analysis. The "Helper LLM" identifies weak paragraphs and proposes concrete actions to transform a mere literature listing into a nuanced, insightful academic discussion.

3. **Citation Feedback** (Factual Grounding): To combat the pervasive issue of hallucination, we introduce a novel Sliding-Window Natural Language Inference (NLI) mechanism. This serves as an automated "fact-checker," verifying that every claim is genuinely supported by its citations. If a claim is unsupported, the system emulates a responsible author by either finding a more appropriate reference or rewriting the claim to align with verifiable facts.

The entire process is governed by a core principle of **monotonic improvement**: a revision is accepted only if it demonstrably improves the survey's quality based on a quantitative rubric. This transforms the writing process into a guided, systematic ascent towards higher quality. Our experimental results provide compelling evidence for this new paradigm. As shown in Figure 1, FIKSurvey not only outperforms previous pipeline-based methods but, surpasses the human-authored baseline in key metrics like Synthesis and Critical Analysis. This suggests that our automated peer review loop does not just fix errors but can actively raise the quality ceiling.

This work offers three primary contributions, reflecting a deeper vision for the future of AI-powered academic writing:

1. **A New Philosophy for AI-Powered Academic Writing:** We propose that the path to high-fidelity automated writing lies not in enhancing raw generative power but in **automating the iterative process of critical review and refinement**. FIKSurvey is the first concrete realization of this philosophy.

2. **A Novel Framework with Verifiable Mechanisms:** We provide a robust and practical framework that translates the abstract concept of "peer review" into a set of specific, verifiable algorithms. The Sliding-Window natural language inference (NLI) mechanism for citation validation, in particular, offers a powerful new tool for ensuring the factual integrity of LLM-generated text.

3. **Empirical Proof of a Raised Quality Ceiling and a Vision for Human-AI Collaboration:** Our results demonstrate that an LLM equipped with a self-review mechanism can achieve a level of analytical depth and factual accuracy that was previously unattainable, effectively raising the quality ceiling for automated survey generation. Furthermore, by supporting optional human-in-the-loop feedback, FIKSurvey provides a blueprint for a future where human experts provide high-level strategic guidance, while AI handles the intensive, iterative work of refinement, maximizing both quality and efficiency.

## 2 RELATED WORK

**Long-term Text Generation.** A challenge for LLMs is the ability to effectively process and generate long-form text while maintaining coherence and logical flow over extended passages. One

direct approach to address this is to extend the model's context window. Several works have explored different Positional Encoding (PE) techniques to achieve this (Zaheer et al., 2020; Su et al., 2024; Choromanski et al., 2020). However, a drawback of modifying PE strategies is that it typically requires retraining the model from scratch, which is computationally expensive and resource-intensive (Kaplan et al., 2020). Another line of research focuses on memory-augmented techniques, which avoid architectural retraining (Rae et al., 2019; Wu et al., 2016; Bulatov et al., 2022). For instance, Temp-Lora embeds context information into a temporary LoRA module that is progressively updated during generation (Hu et al., 2022), rather than relying on an extensive context window. While these methods effectively establish relationships among tokens and maintain contextual understanding, they often face the issue of long generation times due to their iterative nature (Katharopoulos et al., 2020; Choromanski et al., 2020). To further accelerate the generation process, hierarchical modeling techniques have been explored extensively (Pappagari et al., 2019; Ainslie et al., 2020). This paradigm captures the inherent hierarchical nature of long-form text by breaking the task into smaller, more manageable parts (Wu et al., 2021). AutoSurvey similarly uses a hierarchical generation paradigm but enhances it by first creating a well-organized outline for guidance (Wang et al., 2024c). Subsequently, it refines the generated content to improve the quality (Lewis et al., 2019). This combination of structured guidance and refinement helps to mitigate the risk of losing long-range dependencies and improves the final output (Zeng et al., 2022).

**Automatic Survey Generation.** Early explorations have primarily focused on constructing systematic generation pipelines (Yao et al., 2023a;b). For instance, AutoSurvey (Wang et al., 2024c) proposed a pioneering four-stage framework encompassing initial retrieval, outline generation, parallel subsection drafting, and integration with evaluation (Wang et al., 2022). By employing a segmented generation process followed by revision, this approach effectively addresses the context window limitations of LLMs and establishes a preliminary quality assessment framework for automated surveys (Brown et al., 2020). Building on this, SurveyX (Liang et al., 2025) further refined the pipeline by decomposing it into 'Preparation' and 'Generation' phases. SurveyX introduced online reference retrieval, a reference preprocessing method termed 'AttributeTree', and a more sophisticated outline optimization strategy, which significantly enhanced the content depth and citation quality of generated surveys while also incorporating diverse formats like figures and tables (Touvron et al., 2023; Ouyang et al., 2022; Chowdhery et al., 2023).

## 3 METHOD

### 3.1 PIPELINE OVERVIEW

Our proposed FIKSurvey supports two settings: (1) the from-scratch setting, where surveys are produced directly from a topic description, and (2) the from-draft setting, where existing drafts are iteratively improved. Both settings follow the same pipeline, with the from-draft workflow cutting off a corner. Two complementary LLMs are employed: a writer LLM that generates and revises content, and a helper LLM that provides evaluation and targeted feedback. Optional human-in-the-loop review can be incorporated when user-specific requirements must be met, enabling tailored adjustments that go beyond generic automated feedback.

As shown in Figure 2, FIKSurvey contains four stages: (1) candidate paper preparation (2) outline generation (3) survey drafting and (4) survey improvement. Details will be described in the subsequent paragraphs.

**Candidate Paper Preparation.** Given a topic and its corpus, FIKSurvey constructs attribute trees to represent candidate papers in a structured and queryable form. Following SurveyX (Liang et al., 2025), each paper is first categorized into one of four types, method, theory, benchmark, or survey. A type-specific template then prompts the LLM to extract salient dimensions from the full text (e.g., background, metrics) from the full text. The results are serialized into JSON-based attribute trees, which serve as compact yet information-rich retrieval units.

**Outline Generation.** Based on the attribute trees, the writer LLM first produces aprimary outline that lays out the top-level sections. The helper LLM critiques coverage, granularity, and title specificity, and issues actionable edits. The writer applies revisions until gains plateau or a cap is reached. Conditioned on the revised primary outline, the writer then generates a secondary outline

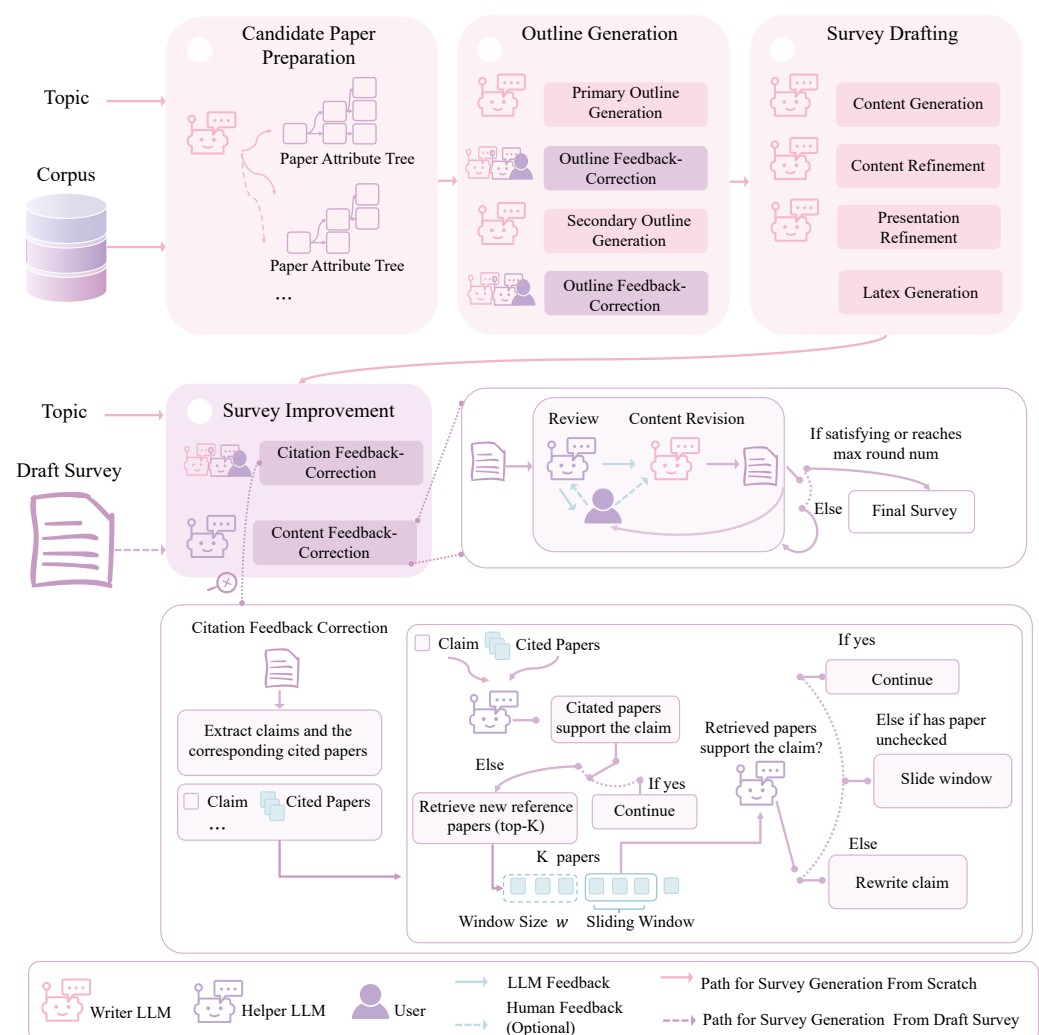

Figure 2: Overview of the FIKSurvey pipeline. FIKSurvey supports both from-scratch and from-draft paths. It integrates automatic feedback across three dimensions: outline, content, and citation, while allowing optional lightweight human feedback. The bottom inset illustrates citation-feedback correction: for each claim–citation pair, existing references are checked; if unsupported, top-$K$ candidate papers are retrieved and validated with a sliding window of size $w$. When support is found, citations are replaced, otherwise the claim can be rewritten.

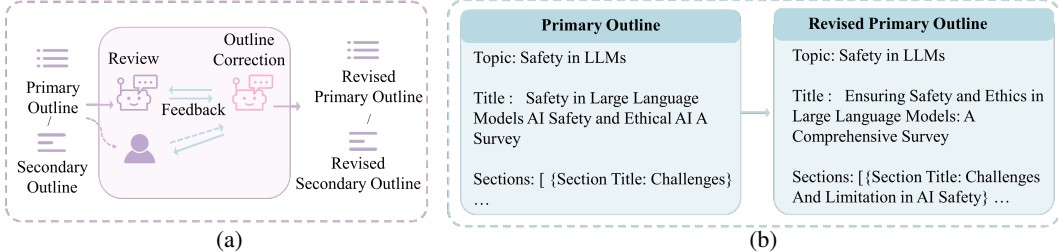

Figure 3: Outline feedback in FiKSurvey. (a) Illustration of outline feedback-correction: the writer LLM generates either a primary or a secondary outline, which is reviewed by the helper LLM. Feedback is incorporated iteratively until the revised outline is finalized. (b) Comparison of the primary outlines before and after outline-feedback correction for topic "Safety in LLMs".

with subsection-level structure and finer decomposition, followed by the same feedback–correction loop (again capped). The final secondary outline is frozen and used as the scaffold for drafting.

**Algorithm: Content Feedback-Correction**

**Inputs:** draft $D$, rubric $\mathcal{M}$, retrieval budget $K$, window size $w$, max rounds $R$.
**Loop (up to $R$ rounds):**

1. **Diagnose.** Helper LLM scores $D$ on rubric dimensions, then identifies weak paragraphs and generates action plans (e.g., enhance paragraph, restructure).

2. **(If needed) Retrieval.** For actions requiring external evidence:

   (a) Retrieve candidate passages (top–$K$).

   (b) **Sliding window over top–$K$:** with window size $w$, move sequentially ($i = 1, 1+w, 1+2w, \dots$):

      i. Form window $\mathcal{W} = \{r_i, \dots, r_{\min(i+w-1, K)}\}$.

      ii. Apply natural language inference (NLI) to check whether any passage in $\mathcal{W}$ supports the target paragraph.

      iii. Stop early if supporting evidence is found.

3. **Edit.** Writer LLM integrates the planned changes and evidence into edits.

4. **Accept/rollback.** Accept only if rubric scores improve; otherwise, revert. Terminate when $R$ rounds are reached or no improvements are possible.

**Output:** revised survey draft $D'$ (retaining only edits with net rubric gains).

**Algorithm: Citation Feedback-Correction**

**Inputs:** draft $D$, claims $C$, current citations $\mathcal{R}$, retrieval budget $K$, window size $w$.
**For each claim $c \in C$:**

1. **Check current citations.** Use NLI to verify which items in $\mathcal{R}(c)$ support $c$. If at least one is supported, keep only the supported subset and stop.

2. **If none support:** retrieve top–$K$ candidate references $\{r_1, \dots, r_K\}$.

3. **Sliding window search.** With window size $w$, move sequentially ($i = 1, 1+w, 1+2w, \dots$):

   (a) Form window $\mathcal{W} = \{r_i, \dots, r_{\min(i+w-1, K)}\}$.

   (b) Rank candidates in $\mathcal{W}$ by NLI score with respect to $c$.

   (c) If a supporting reference $r^\star$ is found: replace the original citation and stop.

4. **If still unsupported:** rewrite the claim to align with available evidence.

5. **Minimal pruning.** If both supported and unsupported citations exist, prune only the unsupported ones.

**Output:** revised survey draft $D'$ with verified citations.

Figure 4: Feedback-correction in FiKSurvey. **Left:** Content refinement loop, where weak paragraphs are diagnosed by the helper LLM, optionally supported with retrieved evidence (top–$K$ searched via a sliding window of size $w$), and revised by the writer LLM. **Right:** Citation validation loop, where claims are first checked against current references. If none support, top–$K$ candidates are scanned with a sliding window until support is found; otherwise, the claim is rewritten. The output in both cases is a revised survey draft with improved content and verified citations.

**Survey Drafting.** Using the accepted outline, the writer LLM produces a structured draft containing sections, subsections, and preliminary citations.

**Survey Improvement.** Drafts are refined through two dedicated modules: (1) Content feedback correction, which enhances clarity, depth, and synthesis; and (2) Citation feedback correction, which enforces factual consistency via claim–citation verification. Workflows of the two modules are provided in Figure 4 and will be detailed in Section 3.2.

## 3.2 FEEDBACK MECHANISM

FIKSurvey systematically injects feedback at three complementary levels: outline, content and citation (factual grounding). Outline level and content level are rubric-driven: each iteration is scored along five dimensions (coverage, structure, relevance, synthesis, critical analysis). A revision is accepted only if the rubric score improves. Otherwise we revert to the previous version, preventing error accumulation and ensuring monotonic improvement. In contrast, citation feedback follows a verification-and-repair loop.

**Outline Feedback.** We cast the outline generation of FIKSurvey as a feedback loop between the writer LLM and the helper LLM. The writer LLM generates a primary and secondary outline, while

the helper LLM evaluates them on rubric dimensions and produces structured feedback. The writer then applies targeted modifications. Revisions are accepted only if scores improve, otherwise the system reverts to the best prior version with a recorded degradation analysis. As shown in Figure 3, the title and section title of the initial primary outline can be revised into more academically appropriate forms. The process iterates up to a configurable maximum number of rounds, ensuring monotonic improvement without excessive drift. More details are provided in Appendix C.1.

**Content Feedback.** As shown in Figure 4 (left), the helper LLM first evaluates the current draft survey against rubric dimensions. (More details are provided in Appendix C.2.) Based on this assessment, it identifies weak paragraphs and assigns targeted actions, and proposes targeted revision strategies, for example, enhancing clarity, restructuring the logical flow, or adding contrasting perspectives (internally mapped to action tags). If external evidence is required, FiKSurvey triggers retrieval-augmented generation (RAG): candidate passages are retrieved (top-$K$) and reranked using a sliding-window *natural language inference (NLI)* model to test entailment between claims and evidence. The top-ranked snippets are then supplied to the writer LLM, which performs LaTeX-aware edits. A revision is accepted only if rubric scores improve, with at most two correction rounds allowed per draft.

**Citation Feedback.** As illustrated in Figure 4 (right), citation validation begins by checking whether the current references already support the claim using sliding-window NLI with window size $w$. If at least one citation is verified, unsupported references are pruned while valid ones are retained. If no support is found, FiKSurvey retrieves the top-$K$ candidate references and slides the window across them to search for supporting evidence. If a valid reference is identified, it replaces the original citation. Otherwise, the system rewrites the claim to align with verifiable evidence. The process ensures that the revised survey maintains only grounded and verifiable citations.

**Optional Human Feedback** FiKSurvey is primarily designed for fully automated survey generation and refinement. Human-in-the-loop review is considered only when users have specific requirements that cannot be satisfied through automation alone. In such cases, minimal expert input can be incorporated to tailor the survey to user needs. Details are provided in Appendix C.4

## 4 EXPERIMENTS

### 4.1 SETUP

#### 4.1.1 FIKSURVEY CONFIGURATION

The FIKSurvey framework supports both generating surveys from scratch and improving existing drafts. It employs two LLMs: a primary writer (GPT-4o) for drafting/refining, and a helper model (GPT-4o-mini) for feedback. It incorporates feedback through three dedicated modules: *outline feedback correction*, *content feedback correction*, and *citation feedback correction*. For outline feedback correction, we allow up to three conversation rounds; for content feedback correction, we limit the process to two rounds; and for citation feedback correction, we adopt a top-$K$ retrieval strategy with a sliding window natural language inference (NLI) to check citation validity, where $K = 10$ and the window size is set to 2.

#### 4.1.2 EVALUATION METRICS

**Content Quality.** Following AutoSurvey (Wang et al., 2024d) and SurveyX (Liang et al., 2025), we evaluate the content quality of generated surveys across five dimensions: (1) *Coverage*:Assesses the comprehensiveness in covering the core and frontier aspects of the topic. (2) *Structure*: Evaluates the logical organization and coherence between sections. (3) *Relevance*: Measures whether the content is focused and free of irrelevant information. (4) *Synthesis*: Assesses the ability to connect different studies into a coherent narrative, rather than a simple enumeration. (5) *Critical analysis*: Evaluates the depth of critical thinking regarding the limitations, contradictions, and open research gaps of existing work. Each dimension is scored on a 1 - 5 scale by an evaluator LLM, using rubric-based prompts adapted from (Wang et al., 2024d; Liang et al., 2025).

**Citation Quality.** To assess factual grounding, we adopt citation *recall* and *precision* metrics. Given a set of claims extracted from the generated survey, we check whether each claim is supported by at least one cited reference (recall), and whether each cited reference is indeed relevant to the claim (precision). Formally,

$$\text{Recall} = \frac{\#\{\text{supported claims}\}}{\#\{\text{all claims}\}}, \quad \text{Precision} = \frac{\#\{\text{supported claim-source pairs}\}}{\#\{\text{all claim-source pairs}\}}.$$

Support is determined via an entailment-based classifier that evaluates whether the core of a claim can be inferred from the cited abstract. This aligns with the implementation in AutoSurvey (Wang et al., 2024d) and SurveyX (Liang et al., 2025). Because hallucinated claims often manifest as unsupported or irrelevant citations, improvements in citation quality directly indicate reductions in hallucination, thereby increasing the reliability of the generated surveys.

### 4.1.3 BASELINES

We compare FIKSurvey against three baselines: (1) AutoSurvey: A systematic framework for automated survey generation. (2) SurveyX: A latest method which use attribute trees for evidence-driven writing to reduce hallucination. (3) Human: Published, human-authored surveys from arXiv.

## 4.2 RESULTS AND ANALYSIS

### 4.2.1 QUANTITATIVE RESULTS

We first evaluate FiKSurvey in the from-scratch generation setting. As shown in Table 1, FiKSurvey achieves the best overall performance among automated methods and produces results comparable to the human baseline. Notably, FiKSurvey can achieve relatively good results on *Synthesis* and *Critical Analysis*, while maintaining high citation quality, demonstrating the effectiveness of its feedback-driven mechanism in uncovering deeper connections across studies and fostering critical yet reliable insights. We further report the quantitative results of FiKSurvey in the *from-draft* setting, as shown in Table 2. Compared to the initial draft surveys, those refined by FiKSurvey exhibit consistent improvements across all dimensions, with particularly notable gains in citation quality. All reported scores are averaged over 20 topic-specific surveys.

Table 1: Comparison of models on content quality and citation metrics.

| Method | Content | | | | | | Citation | |
| --- | --- | --- | --- | --- | --- | --- | --- | --- |
| | Coverage | Structure | Relevance | Synthesis | Critical Analysis | Avg | Recall (%) | Precision (%) |
| AutoSurvey (Wang et al., 2024d) | 4.73 | 4.33 | 4.86 | 4.00 | 3.73 | 4.33 | 82.25 | 77.41 |
| SurveyX (Liang et al., 2025) | 4.91 | 4.73 | 4.86 | 4.09 | 4.00 | 4.52 | 87.80 | 86.78 |
| FiKSurvey (Ours) | 4.95 | 4.90 | 5.00 | 4.65 | 4.55 | 4.81 | 97.33 | 92.85 |
| Human | 5.00 | 4.95 | 5.00 | 4.44 | 4.38 | 4.75 | 86.33 | 77.78 |

Table 2: Quantitative results of draft surveys and their improved versions refined by FiKSurvey.

| | Content | | | | | | Citation | |
| --- | --- | --- | --- | --- | --- | --- | --- | --- |
| | Coverage | Structure | Relevance | Synthesis | Critical Analysis | Avg | Recall (%) | Precision (%) |
| Draft surveys | 4.90 | 4.75 | 4.81 | 4.10 | 4.00 | 4.52 | 81.22 | 80.62 |
| Improved by FiKSurvey | 4.95 | 4.95 | 4.95 | 4.05 | 4.10 | 4.60 | 97.22 | 95.95 |

### 4.2.2 A CLOSER LOOK AT THE FEEDBACK MECHANISM

In this section, we take a closer look at the role of feedback in FiKSurvey. FiKSurvey incorporates feedback along three complementary dimensions: outline, content, and citation. Each dimension addresses distinct quality concerns, such as structural soundness and factual grounding. To understand their contributions, we conduct ablation studies in two complementary ways: (1) a step-wise ablation where feedback modules are progressively removed (Table 3); and (2) a module-wise ablation where each module is individually excluded while keeping the others intact (Table 4).

**Step-Wise Ablation.** Table 3 shows that removing the *content feedback* module alone already leads to a small drop in synthesis and critical analysis, but citation quality remains largely intact. When

both *content* and *citation* feedback are removed, citation recall and precision fall sharply (from over 90% to around 60–77%), indicating the central role of citation verification in maintaining factual grounding. Finally, removing all three feedback components substantially degrades both content and citation metrics, demonstrating that their cumulative effect is essential for stable quality.

**Module-Wise Ablation.** Table 4 provides a finer-grained view. Here, removing *content feedback* mostly impacts synthesis and critical analysis, while removing *citation feedback* causes the largest drop in citation recall and precision. By contrast, removing *outline feedback* mainly affects coverage and structure but leaves citation quality relatively stable. Taken together, these results highlight that each dimension plays a distinct and complementary role in ensuring the overall survey quality.

**Human Feedback.** Although FiKSurvey is designed to minimize manual intervention, optional human feedback can still offer targeted and customized improvements. As illustrated in Figure 5, human reviewers are able to flag vague subsection titles or misaligned content, prompting the writer LLM to generate more precise and well-scoped revisions. This demonstrates how even light-touch human involvement can complement LLM feedback and enhance the final output. Further details are provided in the Appendix.

**Human–LLM Feedback.** Beyond purely human review, FiKSurvey also supports hybrid collaboration during outline generation. Here, human experts provide lightweight guidance (e.g., pointing out that section titles are overly broad), while the helper LLM proposes concrete refinements such as re-worded titles or added subtopics. This division of labor requires only limited expert effort but yields substantially clearer and more academically precise structures. An example of such interaction is shown in Figure 6.

Table 3: Step-wise ablation of the feedback mechanism in FiKSurvey. Feedback components are progressively removed in the order Content → (Content + Citation) → (Content + Citation + Outline). Experiments are conducted on 10 randomly sampled topics. Results show that each feedback dimension contributes to both content and citation quality, and their cumulative effect is essential.

| Method | Content Quality | | | | | | Citation Quality | |
|---|---|---|---|---|---|---|---|---|
| | Coverage | Structure | Relevance | Synthesis | Critical Analysis | Avg | Recall (%) | Precision (%) |
| w/o Content Feedback | 4.90 | 5.00 | 5.00 | 4.70 | 4.30 | 4.78 | 97.26 | 93.46 |
| w/o Content + Citation Feedback | 5.00 | 4.80 | 5.00 | 4.40 | 4.20 | 4.68 | 77.40 | 60.56 |
| w/o Content + Citation + Outline Feedback | 4.90 | 4.70 | 4.90 | 4.20 | 4.10 | 4.78 | 76.20 | 60.91 |
| **FIKSurvey (full)** | 4.90 | 5.00 | 5.00 | 4.80 | 4.50 | 4.84 | 97.20 | 93.38 |

Table 4: Module-wise ablation of feedback in FIKSurvey. We isolate the effect of removing each feedback dimension individually. Content feedback proves crucial for synthesis and critical analysis, citation feedback dominates factual grounding, and outline feedback ensures structural soundness. Together, they provide complementary improvements to survey quality.

| Ablation Module | Content Quality | | | | | | Citation Quality | |
|---|---|---|---|---|---|---|---|---|
| | Coverage | Structure | Relevance | Synthesis | Critical Analysis | Avg | Recall (%) | Precision (%) |
| w/o Content Feedback | 4.90 | 5.00 | 5.00 | 4.70 | 4.30 | 4.78 | 97.26 | 93.46 |
| w/o Citation Feedback | 5.00 | 4.50 | 5.00 | 4.20 | 4.20 | 4.58 | 80.53 | 65.67 |
| w/o Outline Feedback | 4.80 | 4.40 | 5.00 | 4.00 | 4.30 | 4.50 | 97.41 | 93.87 |
| **FIKSurvey (full)** | 4.90 | 5.00 | 5.00 | 4.80 | 4.50 | 4.84 | 97.20 | 93.38 |

## 5 DISCUSSION AND FUTURE WORK

Our FIKSurvey is guided by feedback. Therefore, designing feedback is crucial. To assess the rationality of our design, we analyzed 25 survey submissions and 183 associated reviews from Open-Review (2021–2025), yielding 564 review sentences. Each sentence was categorized into five key dimensions, coverage, structure, relevance, synthesis, and critical analysis. Representative snippets were then supplied to multiple LLMs, which engaged in multi-round discussion. The analysis shows that these five dimensions are repeatedly highlighted by reviewers as central to survey quality, which validates that our feedback captures the concerns of human experts (Details in Appendix E).

Several insights emerge. First, while equal weighting across dimensions offers a balanced baseline, review evidence indicates that salience may vary by context: coverage and relevance dominate

---

**Case Study (Topic: LLM-based Agents)**

**Draft Survey (Excerpt).**
*2 Background and Definitions*
- *2.4 Evolution and Advancements in LLMs*
- *2.5 Integration into AI Agent*
  *"Integrating large language models (LLMs) into artificial intelligence (AI) ..."*
  . . .

**Human Review.**
"The subsection title is ambiguous. What is being integrated? Please revise the title and align the content accordingly. Be more structured."

**Writer LLM Plan.**
Clarify *what* is integrated; restructure by **Architecture / Functionality / Implications**. (No RAG required; structural clarity rewrite.)

**Rewritten Result.**
*"2.5 Integration of Large Language Models into AI Agents.*
Large Language Models (LLMs) have emerged as pivotal components in the development of advanced AI agents..."

Figure 5: Case study of human feedback improving draft survey quality. We show how a vague draft subsection title and content (2.5) is revised after human review, with the writer LLM producing a more clarified version.

---

**Conversation Snapshot (Human – Helper LLM)**

*Human:* Give me some suggestions on refining the section title.

*Helper LLM:* Refining section titles can significantly enhance clarity. Suggestions:
- **Background and Definitions** $\rightarrow$ *Foundations and Evolution: Defining Key Concepts in Machine Learning*
- **In-Context Learning** $\rightarrow$ *In-Context Learning: Mechanisms, Benefits, and Challenges*
- **Few-Shot Learning** $\rightarrow$ *Few-Shot Learning: Bridging the Gap with Minimal Data*
- **Contextual Understanding** $\rightarrow$ *Contextual Understanding: Enhancing Model Comprehension and Interpretation*
- **Transfer Learning** $\rightarrow$ *Transfer Learning: Leveraging Pre-trained Models for New Horizons ...*

**Outline Comparison (Excerpt)**

**Original Draft Outline.**
- Introduction
- Background and Definitions
- In-Context Learning
- Few-Shot Learning
  ...
- Conclusion

**Final Draft Survey Outline.**
- *Introduction*
- *Foundations and Evolution: Defining Key Concepts in Machine Learning*
- *In-Context Learning: Mechanisms, Benefits, and Challenges Few-Shot Learning: Bridging the Gap with Minimal Data*
  ...
- Conclusion

Figure 6: Case study of human–LLM feedback for outline refinement. **Left**: conversation snapshot of helper LLM suggestions. **Right**: comparison of original outline and the final refined version.

in fast-evolving fields, whereas synthesis and critical analysis are emphasized in more mature areas. This suggests that **adaptive weighting**—conditioned on venue or topic dynamics—may yield more precise guidance than static weights. Second, reviewers frequently intertwine criteria, such as linking inadequate coverage to weak synthesis, which motivates **modeling inter-dimensional interactions**. Third, discrepancies between reviewer critiques and LLM judgments highlight the need for **uncertainty-aware feedback**, where system confidence modulates the strength of intervention.

Looking ahead, FIKSurvey can be extended along three promising directions. One is to embed **context-sensitive feedback**, dynamically adapting evaluation rubrics to domain-specific priorities. Another is to incorporate **reviewer–LLM agreement signals**, using divergence as a trigger for more intensive analysis. A third is to move beyond lightweight correction toward **reasoning-driven feedback loops**, in which LLMs not only flag deficiencies but also suggest structural revisions or novel organizational perspectives.

## 6 ETHICS STATEMENT

All authors of this paper have read and agree to abide by the ICLR Code of Ethics. We confirm that this research adheres to the principles set forth in the Code, from its conception and execution to the reporting of its results.

Specifically, we are committed to the following principles:

- **Contribute to society and to human well-being:** This research aims to make a positive contribution to the field of LLM. We have carefully considered its potential societal impacts and have striven to minimize any negative consequences.
- **Uphold high standards of scientific excellence:** We commit to transparency in our research methods, authenticity of our data, and accuracy in reporting our results. We have presented our work in a manner that supports reproducibility.
- **Avoid harm:** We have assessed and worked to mitigate potential harms that could arise from our research, including impacts on individual privacy, data security, and social equity.
- **Respect the work required to produce new ideas and artefacts:** We have given full respect and appropriate credit to the research contributions of others through proper citation.

This statement applies to all activities related to this conference, including the submission, review process, and subsequent academic discussions. We understand and agree that any violation of the ICLR Code of Ethics may lead to the rejection of our submission. We pledge to adhere to these ethical principles throughout the conference and in our future academic activities.

## 7 REPRODUCIBILITY STATEMENT

We are committed to ensuring the full reproducibility of our research. To this end, we provide detailed descriptions of our framework design, algorithm implementations, experimental configurations, and key resources within the main body of the paper and its appendices. This is intended to support our peers in verifying, reproducing, and extending our work.

- **Framework and Algorithms:** The complete pipeline of our core contribution, the FIKSurvey framework, is detailed in Figure 2. The pseudocode for our key feedback mechanisms, including the content feedback-correction and citation feedback-correction algorithms, is provided in Figure 4. Detailed workflows and design principles for the outline, content, and citation feedback mechanisms can be found in Appendix C.1, C.2, and C.3, respectively.
- **Models and Configuration:** Our experiments utilize two Large Language Models: GPT-4o as the "Writer" and GPT-4o-mini as the "Helper". All Retrieval-Augmented Generation (RAG) components use the `bge-base-en-v1.5` embedding model. Key hyperparameter settings, such as the maximum number of rounds for outline feedback (3) and content feedback (2), as well as the retrieval budget ($K = 10$) and sliding window size ($w = 2$) for citation feedback, are explicitly stated in Section 4.1.1 and Appendix B.
- **Prompts:** To ensure a transparent and reproducible feedback loop, we provide the core prompt templates used to drive the "Helper" LLM's feedback generation. Specifically, prompt templates for outline feedback are available in Appendix C.1.2, and those for content feedback are in Appendix C.2.2. These prompts are critical for replicating our feedback mechanism.
- **Datasets and Corpus:** Our experiments are based on 20 public academic topics, which are listed in Table 5 in Appendix A. We directly adopt the corresponding corpus provided by SurveyX (Liang et al., 2025) and follow its procedure to construct attribute trees for the papers, as noted in Appendix B.
- **Source Code:** To maximize support for community reproduction and follow-up research, we commit to open-sourcing the complete implementation source code, experimental scripts, and relevant configuration files on a public code hosting platform (e.g., GitHub) upon acceptance of the paper.

We believe that by combining the detailed methodology, pseudocode, hyperparameter configurations, and prompt templates provided in this paper, along with the forthcoming open-source code, our experimental results can be reliably reproduced.

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

## A  SURVEY TOPICS

Table 5 lists the 20 survey topics used in our main evaluation experiments. For reference, we also provide the titles of the corresponding surveys authored by human experts.

Table 5: Survey topics and corresponding human-written survey titles.

| Topic | Survey Title |
|---|---|
| In-context Learning | A Survey for In-context Learning |
| LLMs for Recommendation | A Survey on Large Language Models for Recommendation |
| LLM-Generated Texts Detection | A Survey of Detecting LLM-Generated Texts |
| Explainability for LLMs | Explainability for Large Language Models |
| Evaluation of LLMs | A Survey on Evaluation of Large Language Models |
| LLMs-based Agents | A Survey on Large Language Model based Autonomous Agents |
| LLMs in Medicine | A Survey of Large Language Models in Medicine |
| Domain Specialization of LLMs | Domain Specialization as the Key to Make Large Language Models Disruptive |
| Challenges of LLMs in Education | Practical and Ethical Challenges of Large Language Models in Education |
| Alignment of LLMs | Aligning Large Language Models with Human |
| ChatGPT | A Survey on ChatGPT and Beyond |
| Instruction Tuning for LLMs | Instruction Tuning for Large Language Models |
| LLMs for Information Retrieval | Large Language Models for Information Retrieval |
| Safety in LLMs | Towards Safer Generative Language Models: Safety Risks, Evaluations, and Improvements |
| Chain of Thought | A Survey of Chain of Thought Reasoning |
| Hallucination in LLMs | A Survey on Hallucination in Large Language Models |
| Bias and Fairness in LLMs | Bias and Fairness in Large Language Models |
| Large Multi-Modal Language Models | Large-scale Multi-Modal Pre-trained Models |
| Acceleration for LLMs | A Survey on Model Compression and Acceleration for Pretrained Language Models |
| LLMs for Software Engineering | Large Language Models for Software Engineering |

## B  IMPLEMENTATION DETAILS

In FIKSurvey, all RAG components use `bge-base-en-v1.5` (Zhang et al., 2024) as the embedding model. For both outline- and content-feedback correction, the framework supports iterative multi-round refinement, with the maximum number of rounds set to 3 and 2, respectively. Since our focus is not on literature collection itself, for the selected 20 topics we directly adopt the corpus

provided by SurveyX (Liang et al., 2025) and follow its procedure to construct attribute trees for candidate papers.

## C  DETAILS OF FEEDBACK MECHANISM

### C.1  DETAILS OF OUTLINE FEEDBACK

We treat survey outline construction as a closed-loop control problem. The helper LLM produces structured, machine-readable feedback, then the writer LLM applies targeted revisions under explicit constraints. An acceptance gate decides whether to keep or rollback based on quality deltas. This design converts vague reviewer-style "comments" into actionable signals that drive monotonic improvement.

#### C.1.1  WORKFLOW OF OUTLINE-FEEDBACK MECHANISM IN FIKSURVEY

**Steps**. Our pipeline proceeds in seven steps:

1. Primary outline generation (writer LLM).

2. Primary outline review (helper LLM).

3. Primary modification loop (writer LLM $\leftrightarrow$ helper LLM, rollback if degraded).

4. Paper mounting and clue extraction for secondary outline generation.

5. Secondary outline generation (writer LLM).

6. Secondary outline review and modification loop (helper LLM $\leftrightarrow$ writer LLM, rollback).

7. Finalization and logging (best version, JSON export).

Paper mounting is a preparatory step that grounds the secondary outline in concrete evidence. Each candidate paper is assigned to a primary outline section by matching its title or abstract, and concise clues are extracted (e.g., datasets, benchmarks, method families, limitations). These clues provide the Writer LLM with domain-grounded anchors when expanding the primary sections into more detailed secondary subsections, thereby improving coverage and reducing superficial enumeration.

**Feedback Artifacts**. The loop exchanges four types of signals:

- Review signals (Primary Outline): lenient 1–5 scores across five dimensions, accompanied by short `notes` (one–two sentence comments clarifying why a score was given) and `quick_wins` (low-cost, high-impact edits such as renaming an ambiguous section title).
- Review signals (Secondary Outline): similar output to review signals for primary outline.
- Prescriptive signals: modification instructions that translate evaluation into concrete edits, while respecting stage-specific constraints.
- Diagnostic signals: degradation analysis that explains why a revision reduced quality and suggests alternative strategies for the next iteration.

**Control Logic.** We accept a revision if its average score improves or meets a threshold; otherwise we rollback to the best-known version. Primary outlines preserve the number and order of sections. Secondary outlines allow flexible restructuring of subsections. This separation stabilizes global structure while enabling local refinement.

#### C.1.2  PROMPT TEMPLATES

We include below the excerpts of representative prompts used in the feedback loop.

**Prompt: Build Feedback for Primary Outline (Review)**

You are given the PRIMARY OUTLINE (headings only) of a survey on "{topic}".
```
<OUTLINE>
{outline_text}
</OUTLINE>
```

Goal: Provide a LENIENT outline-only assessment on five dimensions: coverage, structure, relevance, synthesis, critical analysis.
Guidelines (be flexible):
- Count synonyms/nearby headings as present (e.g., "Landscape/Map/Taxonomy").
- Partial presence is OK; give credit if the intent is visible.
- If unsure, lean slightly positive.

Soft signals to consider:
- coverage: landscape/taxonomy, datasets/benchmarks, method families, scope
- structure: tidy hierarchy
- synthesis: unifying section or hint ("framework/axes/overview").
- critical_analysis: limitations, open problems, threats.   Scoring (1–5, lenient):  1 = mostly missing; 2 = many gaps; 3 = basically there; 4 = mostly there; 5 = very strong.
Return JSON ONLY:

```
{
  "stage": "primary",
  "coverage": <int>, "structure": <int>, "relevance": <int>,
  "synthesis": <int>, "critical_analysis": <int>,
  "notes": {"coverage": "...", "structure": "...", ...},
  "quick_wins": ["short fix 1", "short fix 2"]
}
```

**Prompt: Modification Instructions**

Based on the following outline evaluation, provide specific, actionable modification instructions.
Stage-specific constraints:
- PRIMARY: cannot add/remove/reorder sections; edit titles and descriptions only.
- SECONDARY: can add/remove/reorder subsections.
Return in format:
MODIFICATION INSTRUCTIONS: 1. [specific change]
2. [specific change]
RATIONALE: why changes improve.
EXPECTED IMPROVEMENTS: predicted score changes.
JSON Schema Examples:
Primary:

```
{"title":"Survey Title","sections":[
 {"section title":"...","description":"..."}]}
```

Secondary:

```
{"title":"Survey Title","sections":[
 {"section title":"...","description":"...",
  "subsections":[{"subsection title":"..." ...}]}]}
```

---

**Prompt: Build Feedback for Secondary Outline (Review)**

You are given the SECONDARY OUTLINE (headings with bullets/placeholders) of a survey on "{topic}".
`<OUTLINE>` {outline_text} `</OUTLINE>`
Goal: Provide a LENIENT outline-only assessment on five dimensions: coverage, structure, relevance, synthesis, critical_analysis.
Be flexible: - Credit bullets that indicate intent (assumptions/limits/relations). - Synonyms count; partial coverage still earns points. - If unsure, lean slightly positive.
Soft signals to consider: - coverage: method families, datasets, benchmarks, peripheral topics. - structure: consistent levels, no orphans, occasional mini-summary is a plus. - relevance: bullets on-topic, scoped. - synthesis: unifying/comparison section or relation bullets. - critical_analysis: limits, pitfalls, open problems.
Scoring (1–5, lenient).
Return JSON ONLY:

```
{
  "stage": "secondary",
  "coverage": <int>, "structure": <int>, "relevance": <int>,
  "synthesis": <int>, "critical_analysis": <int>,
  "highlights": ["what works"],
  "gaps": ["missing piece"],
  "next_steps": ["practical tweak"]
}
```

---

**Prompt: Degradation Analysis**

You are analyzing why an outline modification led to a score decrease.
PREVIOUS OUTLINE: ... (score = X)
MODIFIED OUTLINE: ... (score = Y)
SCORE CHANGE: (negative)
Please analyze:
1. Which changes caused degradation?
2. Which aspects worsened?
3. What should be done differently?

Return:
DEGRADATION ANALYSIS: [detailed analysis]
SUGGESTED ALTERNATIVE APPROACH: [specific next step].

---

## C.2 DETAILS OF CONTENT FEEDBACK

We extend the closed-loop feedback principle from outlines to full survey drafts. In this setting, a helper LLM produces structured, machine-readable review signals (scores, notes, and action tags) at the paragraph level, while the writer LLM applies targeted revisions under explicit constraints. An acceptance gate decides whether to accept or rollback changes depending on quality deltas. When external support is required, retrieval-augmented generation (RAG) with sliding-window NLI ensures that only evidence-entailing passages are integrated, reducing hallucination risk. This design transforms vague reviewer-style comments into actionable, localized edits that improve clarity, logical flow, and critical depth.

### C.2.1 WORKFLOW OF CONTENT-FEEDBACK MECHANISM IN FIKSURVEY

The content-feedback pipeline proceeds in four steps:

1. Content review (helper LLM). Score the draft on five rubric dimensions; provide short notes and assign action tags to weak paragraphs.

2. Modification instructions (helper LLM). Translate action tags into concrete editing suggestions.

3. Revision (writer LLM). Apply paragraph-level edits in LaTeX, optionally using retrieved evidence (top-$K$ snippets verified via NLI).

4. Acceptance gate. Re-evaluate the revised draft; accept if scores improve, rollback otherwise. A maximum of two correction rounds are allowed per draft.

### C.2.2 PROMPTS FOR CONTENT FEEDBACK

In FIKSurvey, content refinement relies on structured prompts to ensure that evaluations, modification plans, and RAG-enhanced revisions remain machine-readable and auditable. Below we reproduce the core prompts.

---

**Prompt: Review (Coverage)**

You are a reviewer evaluating an academic survey draft.
Dimension: COVERAGE
Task: Assess how well the draft survey covers the breadth of the field. Does it include central methods, datasets, benchmarks, and peripheral topics? Identify major gaps or missing areas.
Scoring (1–5): 1 = very poor coverage; 5 = very comprehensive.
Return JSON only:

```
{
   "dimension": "coverage",
   "score": <1-5>,
   "reasoning": "...",
   "improvement_plan": "..."
}
```

---

**Prompt: Review (Structure)**

You are a reviewer evaluating an academic survey draft.
Dimension: STRUCTURE
Task: Assess the logical organization of the survey. Is the hierarchy clear and consistent? Does the flow make sense (e.g., Problem/Scope → Landscape → Methods → Synthesis → Future)?
Scoring (1–5): 1 = very disorganized; 5 = clear and coherent.
Return JSON only:

```
{
   "dimension": "structure",
   "score": <1-5>,
   "reasoning": "...",
   "improvement_plan": "..."
}
```

---

**Prompt: Review (Relevance)**

You are a reviewer evaluating an academic survey draft.
Dimension: RELEVANCE
Task: Assess whether the survey content is on-topic and aligned with the intended scope. Are tangential or irrelevant areas included? Are exclusions or boundaries clearly stated?
Scoring (1–5): 1 = mostly irrelevant; 5 = highly focused and on-topic.
Return JSON only:

```
{
   "dimension": "relevance",
   "score": <1-5>,
   "reasoning": "...",
   "improvement_plan": "..."
}
```

**Prompt: Review (Synthesis)**

You are a reviewer evaluating an academic survey draft.
Dimension: SYNTHESIS
Task: Assess whether the survey synthesizes and compares works, rather than just listing them.
Does it provide unifying frameworks, axes, or comparative insights?
Scoring (1–5): 1 = no synthesis, just enumeration; 5 = strong unifying synthesis.
Return JSON only:

```
{
  "dimension": "synthesis",
  "score": <1-5>,
  "reasoning": "...",
  "improvement_plan": "..."
}
```

**Prompt: Review (Critical Analysis)**

You are a reviewer evaluating an academic survey draft.
Dimension: CRITICAL ANALYSIS
Task: Assess whether the survey includes critical perspectives. Does it discuss limitations, open
problems, threats to validity, or future challenges?
Scoring (1–5): 1 = no critical analysis; 5 = strong and insightful critique.
Return JSON only:

```
{
  "dimension": "critical_analysis",
  "score": <1-5>,
  "reasoning": "...",
  "improvement_plan": "..."
}
```

**Prompt: Parse Improvement Plan**

You are a technical editor analyzing survey improvement suggestions.
Parse the following improvement plan and extract specific, actionable modifications in JSON
format.
Dimension: {dimension} Improvement Plan: {improvement_plan}
For each suggested modification, return JSON with this structure:

```
{
    "type": "add_section|insert_subsection|add_table|
    add_timeline|enhance_paragraph",
    "target_location": "section_name or general_location",
    "description": "specific description of what to do",
    "content_keywords": ["key", "terms", "to", "search"],
}
```

Return only a JSON array of modification objects.

---

**Prompt: Generate Enhanced Content with RAG**

You are a technical writer creating content for an academic survey.
Based on the following action description and reference materials, generate appropriate LaTeX content.
Action Type: {action.action_type}
Action Description: {action.description}
Reference Materials: {references_text}
Requirements:
1. Generate content appropriate for the action type:
- add_section: Full section with \section{} command and content
- insert_subsection: Full subsection with \subsection{} command and content
- add_table: LaTeX table with \begin{table} environment
- add_timeline: Structured chronological content or table
- enhance_paragraph: Detailed paragraph content
2. Include proper LaTeX citations using \cite{} for the references
3. Use academic writing style appropriate for surveys
4. Make content substantive and well-structured
5. Ensure content is directly relevant to the action description
Generated LaTeX content.

---

## C.3  DETAILS OF CITATION FEEDBACK

### C.3.1  WORKFLOW OF CITATION-FEEDBACK MECHANISM IN FIKSURVEY

The process of citation-feedback mechanism can be summarized in the following steps:

1. Locate cited sentences. Identify sentences containing citations and split them into the claim text and its citation keys.

2. Check support. For each claim–citation pair, retrieve the cited paper and test whether the claim is supported using a natural language inference (NLI) model.

3. Handle supported claims. If support is found, optionally prune away unsupported references and keep only the valid ones.

4. Diagnose unsupported claims. If no support is found, a helper LLM generates a short diagnosis explaining why the claim fails.

5. Propose fixes. The helper LLM suggests a factual rewrite, style notes, and alternative phrasings in a structured format.

6. Retrieve candidate references. Search for new candidate citations from the corpus and verify them with the NLI model.

7. Apply repair strategies. Depending on configuration, repair may prioritize adjusting references, rewriting the claim, or both. A revision is accepted only if it is NLI-supported.

8. Finalize and log. The corrected LaTeX file and a JSON record of diagnostics, fixes, and chosen actions are saved for transparency.

This workflow turns judgments of the citations into explicit, machine-verifiable feedback, enabling automatic citation repair in a transparent and auditable way.

## C.4  DETAILS OF HUMAN FEEDBACK

In this module, the writer LLM serves as the central actor: it receives human opinions, structured plans, and optional evidence, and is responsible for carrying out edits. The workflow unfolds as following steps:

1. Opinion Collection. Human feedback items are gathered (ID, category, priority, text). These define the editing objectives.

2. Structure Analysis. The writer LLM summarizes the current draft into JSON, providing contextual grounding for later editing.

3. Global Modification Plan. The writer LLM integrates all opinions into a structured plan specifying execution order, dependencies, and evidence needs.

4. Evidence Provision. For opinions requiring external support, candidate passages are retrieved, filtered via NLI entailment, and supplied as verified evidence.

5. Stepwise Modification. Guided by the plan, the Writer applies LaTeX-aware edits using the draft, the given opinion, and optional evidence, while preserving global structure.

6. Acceptance Gate And Rollback. Each revision is re-scored on five dimensions. If quality degrades, the system reverts to the prior version and logs a degradation note.

7. Finalization And Reporting. The Writer's best-performing version is exported as the final draft, together with a structured log of all modifications.

# D  CASE STUDY OF OPTIONAL HUMAN FEEDBACK

In this section, we provide example of human feedback for content refinement in Figure 7 and example of human-LLM feedback for outline refinement in Figure 8.

---

**Case Study (Topic: LLM-based Agents)**

**Draft Survey (Excerpt).**
*1 Introduction*
- *1.1 Significance of LLMs-based Agents*
  The emergence of large language models (LLMs) has significantly transformed artificial intelligence, particularly in natural language processing (NLP), by enhancing AI agents ...
- *1.2 Objectives of the Survey*
  This survey provides a comprehensive examination of LLM-based agents in natural language processing, focusing on several key objectives. Firstly, it explores the potential benefits and challenges of deploying large language ...
  ...

**Human Review.**
"The introduction lacks sufficient background on recent developments in large language models and their role in autonomous agents."

**Rewritten Result.**
*1 Introduction*
Recent developments in large language models (LLMs) have revolutionized the landscape of Artificial Intelligence and natural language processing (NLP). These models, such as GPT-4 and others, have demonstrated unprecedented performance across an array of language tasks, culminating in human-like capabilities in understanding and generating text. Notably, the advancements in training techniques, scaling parameters, and architectural refinements have accelerated their integration into autonomous agents.
LLMs are now foundational in creating autonomous agents capable of performing complex interactions in real-time environments. Their ability to process and generate natural language allows for sophisticated dialogues, making them adept in various applications ranging from customer service to academic support. The emergence of multimodal frameworks further showcases LLMs' capacity to bridge language and visual data, offering new dimensions to autonomous systems.
Over the past few years, the significance of LLMs in AI has grown substantially, enhancing tasks such as sentiment analysis, information retrieval, and content creation. This paper explores the transformative impact of LLMs-based agents on NLP, emphasizing their recent developments, capabilities, and the challenges that lie ahead.
- *1.1 Significance of LLMs-based Agents*
  The emergence of large language models (LLMs) has significantly transformed artificial intelligence, particularly in natural language processing (NLP), by enhancing AI agents ...
- *1.2 Objectives of the Survey*
  This survey provides a comprehensive examination of LLM-based agents in natural language processing, focusing on several key objectives. Firstly, it explores the potential benefits and challenges of deploying large language ...
  ...

---

Figure 7: Example of human feedback for content refinement.

---

**Conversation Snapshot between Human And Helper LLM (Topic: In-Context Learning)**

*Human:* Give me some suggestions on refining the section title.

*Helper LLM:* Refining section titles can significantly enhance clarity. Suggestions:

- **Background and Definitions** → *Foundations and Evolution: Defining Key Concepts in Machine Learning*
  Rationale: Emphasizes both the foundational definitions and the historical evolution of the concepts.
- **In-Context Learning** → *In-Context Learning: Mechanisms, Benefits, and Challenges*
  Rationale: Clearly outlines the focus on mechanisms, benefits, and challenges, making it easier for readers to know what to expect.
- **Few-Shot Learning** → *Few-Shot Learning: Bridging the Gap with Minimal Data*
  Rationale: Highlights the essence of few-shot learning, which is achieving results with minimal data.
- **Contextual Understanding** → *Contextual Understanding: Enhancing Model Comprehension and Interpretation*
  Rationale: Focuses on the enhancement of comprehension and interpretation, which are key aspects of contextual understanding.
- **Transfer Learning** → *Transfer Learning: Leveraging Pre-trained Models for New Horizons*
  Rationale: Emphasizes the use of pre-trained models to explore new tasks, capturing the essence of transfer learning.
- **Prompt Engineering** → *Prompt Engineering: Crafting Inputs for Optimal Model Performance*
  Rationale: Focuses on the crafting aspect of prompt engineering, which is crucial for optimizing model outputs.

---

**Outline Comparison (Topic: In-Context Learning)**

**Original Draft Outline.**
- Introduction
- Background and Definitions
- In-Context Learning
- Few-Shot Learning
- Contextual Understanding
- Transfer Learning
- Prompt Engineering
- Conclusion

**Final Draft Survey Outline.**
- *Introduction*
- *Foundations and Evolution: Defining Key Concepts in Machine Learning*
- *In-Context Learning: Mechanisms, Benefits, and Challenges*
- *Few-Shot Learning: Bridging the Gap with Minimal Data*
- *Contextual Understanding: Enhancing Model Comprehension and Interpretation*
- *Transfer Learning: Leveraging Pre-trained Models for New Horizons*
- *Prompt Engineering: Crafting Inputs for Optimal Model Performance*
- Conclusion

Figure 8: Example of human–LLM feedback for outline refinement.

## E FURTHER INVESTIGATION ON BUILDING FEEDBACK

To validate the rationality of our feedback design, we collected 25 survey submissions and their associated comments. After filtering the trivial comments, we obtained 183 associated reviews and extracted 564 review sentences. We categorized the extracted sentences into five preliminary dimensions (Coverage, Structure, Relevance, Synthesis, Critical Analysis) using keyword-based bucketing. The result is presented in Figure 9. Representative snippets were then supplied to multiple large language models acting as distinct roles (methodologist, domain expert, statistician). All the roles are implemented by GPT-4o. Through multi-round dialogue, these roles critiqued and refined the rubric; finally, an adjudicator LLM merged the proposals into a unified rubric with normalized weights. This procedure ensured that the resulting framework was not only theoretically motivated but also grounded in real reviewer concerns.

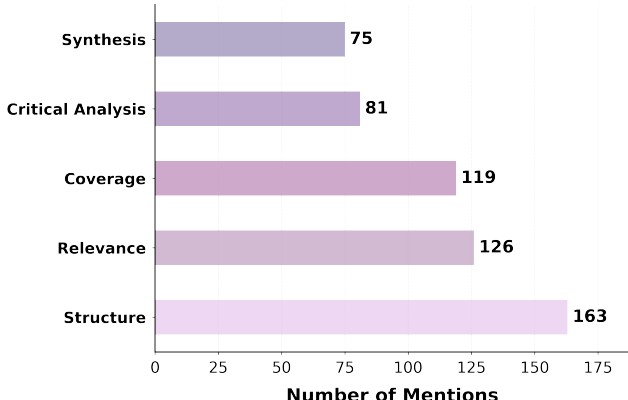

Figure 9: The number of reviewer mentions along the rubric dimension.

**Insights across dimensions.** As shown in Figure 10, the analysis confirms that the five dimensions are strongly supported by recurring reviewer critiques. For example, under *Coverage*, reviewers frequently stressed omissions of recent work or inadequate search protocols. *Structure* was highlighted when papers lacked logical flow or signposting. On *Relevance*, reviewers questioned topical fit and timeliness. For *Synthesis*, concerns centered on superficial taxonomies. Finally, *Critical Analysis* was repeatedly mentioned when surveys lacked balanced critique or articulation of open problems. These consistent patterns suggest that our dimension design is well aligned with human judgment.

**Future directions.** Three directions arise from the review evidence. First, while our current rubric assigns equal weights to the five dimensions, reviewer emphasis clearly varies across domains: Coverage and Relevance dominate in fast-moving fields, whereas Synthesis and Critical Analysis matter more in mature topics. This indicates the need for **adaptive weighting**, conditioned on topic maturity, venue norms, or temporal context. Second, reviewers often noted interdependencies (e.g., weak coverage undermining synthesis). Future feedback mechanisms should therefore **model interactions between dimensions** rather than treating them independently. Third, discrepancies between reviewer critiques and LLM assessments suggest the value of **uncertainty-aware feedback**: when signals are weak or conflicting, the system could modulate its interventions or escalate to optional human feedback. Overall, our analysis demonstrates that the five-dimension rubric is not arbitrary but arises from consistent reviewer expectations. Future work should build on this foundation to design **intelligent, context-sensitive feedback**, extending FIKSurvey from lightweight corrections toward adaptive, reasoning-driven interventions that mirror and augment human reviewing practice.

## F EVALUATION PROMPT

In this section, we provide the prompt we use for evalutation in our experiments.

---

**Report Generated by The Multi-Agent Discussion (Excerpt)**

**Equal Weighting vs. Adaptive Weighting**
The current rubric employs equal weighting for all dimensions, reflecting their collective importance in producing high-quality survey papers. However, this approach may not always capture the nuanced needs of different topics or venues. For instance, some papers may require more emphasis on Synthesis if they aim to introduce new frameworks, while others might prioritize Critical Analysis to challenge existing methodologies. Adaptive weighting, which adjusts the importance of each dimension based on the paper's focus or the venue's expectations, could provide a more tailored evaluation. This approach would allow for a dynamic assessment that better aligns with the specific goals and context of each survey paper.

**Proposing Smarter Constructs**
To enhance the evaluation framework, several smarter constructs could be considered:

1. **Topic/Venue-Adaptive Weights**: Implementing adaptive weights that vary based on the topic or venue could ensure that the evaluation criteria are aligned with the specific objectives and standards of different research areas.

2. **Uncertainty Measures**: Incorporating uncertainty measures could provide insights into the confidence of assessments, particularly in dimensions like Relevance and Critical Analysis where subjective judgments play a significant role.

3. **Reviewer–LLM Agreement**: Analyzing the agreement between human reviewers and language models (LLMs) could offer a novel perspective on the consistency and reliability of evaluations, potentially identifying areas where human judgment diverges from automated assessments.

4. **Interaction Effects**: Exploring interactions between dimensions, such as how Structure influences the effectiveness of Synthesis, could uncover deeper insights into the interdependencies that contribute to a paper's overall quality.

**Research Directions**
Several concrete research directions emerge from this discussion:

- **Developing Adaptive Weighting Systems**: Future research could focus on creating algorithms that dynamically adjust dimension weights based on the paper's topic and venue. This would involve analyzing historical data to identify patterns and preferences specific to different research communities.
- **Integrating Uncertainty Metrics**: Research could explore the development of uncertainty metrics for each dimension, providing evaluators with tools to express confidence levels in their assessments and identify areas of ambiguity.
- **Automating Agreement Analysis**: Investigating the alignment between human and LLM evaluations could lead to the development of automated systems that flag discrepancies and suggest areas for further review.
- **Studying Dimension Interactions**: Conducting empirical studies to examine the interaction effects between dimensions could yield insights into how different aspects of a paper contribute to its perceived quality, informing more holistic evaluation strategies.

In conclusion, while the current rubric provides a solid foundation for evaluating survey papers, there is potential for significant enhancements through adaptive weighting, uncertainty measures, and the exploration of dimension interactions. These advancements could lead to more nuanced and context-sensitive evaluations, ultimately improving the quality and impact of survey papers in the field of machine learning.

Figure 10: Insights reveals by the multi-agent discussion.

## G  STATEMENT ON THE USE OF LARGE LANGUAGE MODELS (LLMS)

In accordance with the conference/journal policy on transparency, we provide a detailed account of the role Large Language Models (LLMs) played in this research. We distinguish between two primary capacities in which LLMs were utilized.

---

**Evaluation Prompt: Content – Coverage**

Here is an academic survey about the topic "{topic}":

---
{content}
---

<instruction> Please evaluate this survey about the topic {topic} based on the criterion provided below, and give a score from 1 to 5 according to the score descriptions.

**Criterion Description (Coverage):** Coverage assesses the extent to which the survey encapsulates all relevant aspects of the topic, ensuring comprehensive discussion on both central and peripheral topics.

**Score Descriptions:**
- **Score 1:** The survey has very limited coverage, only touching on a small portion of the topic and lacking discussion on key areas.
- **Score 2:** The survey covers some parts of the topic but has noticeable omissions, with significant areas either underrepresented or missing.
- **Score 3:** The survey is generally comprehensive in coverage but still misses a few key points that are not fully discussed.
- **Score 4:** The survey covers most key areas of the topic comprehensively, with only very minor topics left out.
- **Score 5:** The survey comprehensively covers all key and peripheral topics, providing detailed discussions and extensive information.

Return the score without any other information.

---

Figure 11: Evaluation prompt for content coverage.

---

**Evaluation Prompt: Content – Structure**

Here is an academic survey about the topic "{topic}":

---
{content}
---

<instruction> Please evaluate this survey about the topic {topic} based on the criterion provided below, and give a score from 1 to 5 according to the score descriptions.

**Criterion Description (Structure):** Structure evaluates the logical organization and coherence of sections and subsections, ensuring that they are logically connected.

**Score Descriptions:**
- **Score 1:** The survey lacks logic, with no clear connections between sections, making it difficult to understand the overall framework.
- **Score 2:** The survey has weak logical flow with some content arranged in a disordered or unreasonable manner.
- **Score 3:** The survey has a generally reasonable logical structure, with most content arranged orderly, though some links and transitions could be improved (e.g., repeated subsections).
- **Score 4:** The survey has good logical consistency, with content well arranged and natural transitions, only slightly rigid in a few parts.
- **Score 5:** The survey is tightly structured and logically clear, with all sections and content arranged most reasonably, and transitions between adjacent sections smooth without redundancy.

Return the score without any other information.

---

Figure 12: Evaluation prompt for content structure.

---

**Evaluation Prompt: Content – Relevance**

Here is an academic survey about the topic "{topic}":

---
{content}
---

<instruction> Please evaluate this survey about the topic {topic} based on the criterion provided below, and give a score from 1 to 5 according to the score descriptions.

**Criterion Description (Relevance):** Relevance measures how well the content of the survey aligns with the research topic and maintains a clear focus.

**Score Descriptions:**
- **Score 1:** The content is outdated or unrelated to the field it purports to review, offering no alignment with the topic.
- **Score 2:** The survey is somewhat on topic but with several digressions; the core subject is evident but not consistently adhered to.
- **Score 3:** The survey is generally on topic, despite a few unrelated details.
- **Score 4:** The survey is mostly on topic and focused; the narrative has a consistent relevance to the core subject with infrequent digressions.
- **Score 5:** The survey is exceptionally focused and entirely on topic; the article is tightly centered on the subject, with every piece of information contributing to a comprehensive understanding of the topic.

Return the score without any other information.

---

Figure 13: Evaluation prompt for content relevance.

---

**Evaluation Prompt: Content – Critical Analysis**

Here is an academic survey about the topic "{topic}":

---
{content}
---

<instruction> Please evaluate this survey about the topic {topic} based on the criterion provided below, and give a score from 1 to 5 according to the score descriptions.

**Criterion Description (Critical Analysis):** Critical Analysis examines the depth of critique applied to existing studies, including identification of methodological limitations, theoretical inconsistencies, and research gaps.

**Score Descriptions:**
- **Score 1:** The survey merely lists existing studies without any analytical commentary or critique.
- **Score 2:** The survey occasionally mentions limitations of studies but lacks systematic analysis or synthesis of gaps.
- **Score 3:** The survey provides sporadic critical evaluations of some studies, but the critique is shallow or inconsistent.
- **Score 4:** The survey systematically critiques most key studies and identifies research gaps, though some areas lack depth.
- **Score 5:** The survey demonstrates rigorous critical analysis of methodologies and theories, clearly maps research frontiers, and proposes novel directions based on synthesized gaps.

Return the score without any other information.

---

Figure 14: Evaluation prompt for content critical analysis.

---

**Evaluation Prompt: Content – Synthesis**

Here is an academic survey about the topic "{topic}":

```
---
{content}
---
```

`<instruction>` Please evaluate this survey about the topic {topic} based on the criterion provided below, and give a score from 1 to 5 according to the score descriptions.

**Criterion Description (Synthesis):** Synthesis evaluates the ability to interconnect disparate studies, identify overarching patterns or contradictions, and construct a cohesive intellectual framework beyond individual summaries.

**Score Descriptions:**
- **Score 1:** The survey is purely a collection of isolated study summaries with no attempt to connect ideas or identify broader trends.
- **Score 2:** The survey occasionally links studies but fails to synthesize them into meaningful patterns; connections are superficial.
- **Score 3:** The survey identifies some thematic relationships between studies but lacks a unified framework to explain their significance.
- **Score 4:** The survey synthesizes most studies into coherent themes or debates, though some connections remain underdeveloped.
- **Score 5:** The survey masterfully integrates studies into a novel framework, revealing latent trends, resolving contradictions, and proposing paradigm-shifting perspectives.

Return the score without any other information.

---

Figure 15: Evaluation prompt for content synthesis.

---

**Evaluation Prompt: Citation**

– Claim: {claim} —
Source: {source} —
Claim: {claim} —
Is the Claim faithful to the Source? A Claim is faithful to the Source if the core part in the Claim can be supported by the Source.
Only reply with 'Yes' or 'No'.

---

Figure 16: Evaluation prompt for citation.

## G.1 LLMs as a Core Component and Subject of the Research Methodology.

The central contribution of this paper, the FIKSurvey framework, is fundamentally designed, built, and operated based on LLMs. Specifically, models such as GPT-4o and GPT-4o mini served as the core technological agents (i.e., the Writer, Helper, and Evaluator) that drive the automated survey generation and the iterative feedback loop. In this context, LLMs were not merely a tool but were the **direct subject of our investigation**. The experimental results presented in this paper are an evaluation of the performance and behavior of these LLM-based agents within our proposed framework. As such, their role was significant and integral to the research itself.

## G.2 LLMs as a General-Purpose Writing Assistance Tool.

During the preparation of this manuscript, the human authors used LLMs (e.g., GPT-4) as a general-purpose assistive tool. This usage was confined to supportive tasks, including: (a) proofreading and stylistic polishing of drafts written by the human authors; (b) rephrasing sentences for improved

clarity and readability; and (c) assisting in generating initial drafts for the abstract and introduction, which were subsequently rewritten and thoroughly revised by the authors.

### G.3 CLARIFICATION ON RESEARCH IDEATION AND AUTHOR RESPONSIBILITY.

It is critical to state that the **conceptualization of the research, the architectural design of the FIKSurvey framework, the experimental design, and the analysis and interpretation of the results were exclusively conducted by the human authors**. The LLMs did not contribute to the research ideation or the formulation of the core scientific concepts.

Ultimately, the **human authors assume full and final responsibility for the entire content of this manuscript**. All text, including passages assisted by LLMs, has been rigorously reviewed, fact-checked, and critically edited by the authors to ensure scientific accuracy, originality, and integrity. We affirm that LLMs do not meet the criteria for authorship and, in line with the stated policy, have not been listed as authors.

