# OpenReview forum: "FIKSurvey: An Automated Peer Review Loop to Raise the Ceiling on AI Academic Writing"
_ICLR.cc/2026/Conference — ICLR 2026 Conference Withdrawn Submission_

### Official Review · Reviewer_4588 · 2025-10-26

**Soundness:** 2
**Presentation:** 2
**Contribution:** 2
**Rating:** 2
**Confidence:** 4

**Summary:**

This paper proposes FIKSurvey, which uses a feedback loop to improve academic writing. The feedback mechanism consists of three dimensions: outline feedback for structural clarity, citation feedback for evidence validation, and content feedback for readability and analytical depth. Experiments show the superiority of this method, outperforming other auto-surveying systems.

**Strengths:**

1. Using LLMs to automatically generate surveys is a highly worthwhile task for exploration.
2. The proposed method outperforms various baselines in the experiments.

**Weaknesses:**

1. The contribution of this paper lies in **prompt engineering**, undermining the innovativeness of this work.
2. The core of the method proposed in this paper is the "feedback mechanism", however **"feedback" is not something really new** in scientific content generation. For example, JRE-L[1] propose a feedback loop that iteratively improve readability of scientific content. I apologize for not listing all relevant papers, but the authors should be able to find more studies that inject feedback into the context to improve scientific writing.
3. The method in this paper contains **numerous if-else components**. Such design and implementation seem inherently to involve **human intervention and engineering design**, which weaken the paper's innovation. Meanwhile, the complex pipeline also potentially undermines reproducibility, which in turn affects the follow-up of future work.
4. The **presentation** of this paper needs improvement. For example, the illustrations of this paper are unclear, like figures 2, 3, and 4. It is hard to understand the workflow and dataflow in Figure 2 and 3, and Figure 4 seems not like either a standard algorithm illustration or illustrative diagram.

[1] Jiang et al. 2025. JRE-L: Journalist, Reader, and Editor LLMs in the Loop for Science Journalism for the General Audience. NAACL 2025.

**Questions:**

1. As a multi-stage prompting method, how do the authors conduct the context management? Are there any challenges and design considerations valuable to the follow-up researchers?
2. Are there any interesting points or uniqueness in the feedback mechanism of this paper, compared with previous work?

---

### Official Review · Reviewer_4rnx · 2025-10-27

**Soundness:** 2
**Presentation:** 3
**Contribution:** 2
**Rating:** 4
**Confidence:** 3

**Summary:**

The paper introduces **FIKSurvey**, a novel framework for automated academic survey generation using Large Language Models (LLMs). Unlike previous **pipeline-based systems** such as **AutoSurvey** and **SurveyX**, which operate sequentially, FIKSurvey adopts a **feedback-driven, closed-loop process** inspired by the human peer-review mechanism.

The system uses:
* **Dual-agent architecture**: Writer LLM (content generation) + Helper LLM (feedback and evaluation).
* **Three-layer feedback loops**:
  * **Outline feedback** – improves survey structure.
  * **Content feedback** – enhances readability, synthesis, and analytical depth.
  * **Citation feedback** – validates citation accuracy using Sliding-Window NLI.

Experimental results across 20 topics show that FIKSurvey outperforms baselines in content quality and citation accuracy.

**Strengths:**

* **Novel closed-loop paradigm:** Introducing peer-review-like iterative refinement is an innovative departure from traditional linear generation frameworks.
* **Effective performance gains:** The framework shows strong improvements over baselines (AutoSurvey, SurveyX) in synthesis, critical analysis, citation recall, and precision.
* **Modular and extensible design:** Supports both survey generation from scratch and refining draft surveys. Optional human-in-the-loop input provides flexibility for domain adaptation.

**Weaknesses:**

* **No cost or efficiency analysis:** The paper does not report computational cost (e.g., GPU hours, API token usage, time per iteration). Given the multi-agent feedback rounds and retrieval steps, this is essential for understanding scalability and accessibility.
* **No human evaluation against baselines:** Although examples of human-in-the-loop refinement are provided, there is no systematic human evaluation comparing outputs from FIKSurvey vs. SurveyX, AutoSurvey, or human-written surveys. The evaluation relies entirely on LLM-based rubric scoring and automated NLI methods.

**Questions:**

- Line 317: Please ensure correct spacing between “Coverage:” and “Assesses”.
- Several occurrences of incorrect or mismatched quotation direction (“ ” vs. “ ”). Please standardize throughout the paper.
- Potential Missing Citation
  * *[Position Paper: How Should We Responsibly Adopt LLMs in the Peer Review Process?](https://openreview.net/forum?id=KZ3NspcpLN)* — Regarding citation verification.
  * *[SurGE: A Benchmark and Evaluation Framework for Scientific Survey Generation](https://arxiv.org/abs/2508.15658)*
  * *[InteractiveSurvey: An LLM-based Personalized and Interactive Survey Paper Generation System](https://arxiv.org/abs/2504.08762)*
  * *[SurveyForge: On the Outline Heuristics, Memory-Driven Generation, and Multi-dimensional Evaluation for Automated Survey Writing](https://arxiv.org/abs/2503.04629)*
  * *[SurveyG: A Multi-Agent LLM Framework with Hierarchical Citation Graph for Automated Survey Generation](https://arxiv.org/abs/2510.07733v1)* — I acknowledge that this paper was released after the submission deadline, so I think just incorporating a discussion would be enough.

---

### Official Review · Reviewer_h7yK · 2025-10-28

**Soundness:** 3
**Presentation:** 2
**Contribution:** 2
**Rating:** 4
**Confidence:** 4

**Summary:**

The paper proposes FIKSurvey, a framework for automated survey generation using dual llm agents (writer and reviewer). The system implements iterative refinement across three dimensions: outline feedback, content feedback, and citation feedback. Results show improvements over AutoSurvey and SurveyX on citation recall and content quality metrics.

**Strengths:**

- The paper tackles an important problem. Automatically generating surveys is an important problem and can greatly accelerate the scientific research process.

- The paper provides thorough ablations. Tables 3-4 provide detailed ablations showing how performance changes when removing each of the three feedback components (outline feedback, content feedback, and citation feedback).

- The paper is generally well-written, with detailed system explanations and prompts provided to ensure reproducibility.

**Weaknesses:**

- The paper has limited novelty. The paper is essentially a complex prompt engineering system, which, while admittedly a useful application, provides little methodological novelty. The multi-agent feedback loop has also been well studied since the paper Reflexion.

- The evaluation mostly relies on human evaluation. I agree that this task is hard to evaluate automatically. However, only 20 examples are evaluated by humans, raising questions about statistical significance and accuracy. The paper also uses LLM-as-judge evaluation, but it is questionable to use GPT-4o to evaluate GPT-4o outputs.

- Only limited baselines (2) on a limited number of topics (20) are compared. The paper would benefit from extending the evaluation beyond AI/ML to other scientific fields and comparing against more baselines in domains such as biology, physics, and social sciences.

**Questions:**

From Tables 3-4, it seems that one of the three components, content feedback, does not contribute significantly to the final content quality and even degrades citation quality. Can the authors explain why this happens and provide some intuition?

---

### Official Review · Reviewer_xGwk · 2025-10-30

**Soundness:** 2
**Presentation:** 3
**Contribution:** 2
**Rating:** 2
**Confidence:** 4

**Summary:**

The paper proposes a framework to automatically generate literature surveys, called FIKSurvey, which incorporates feedback along three dimensions to improve surveys. These dimensions are structural clarity, citation quality, and content readability & depth. The framework incorporates two LLMs that work in tandem: a writer LLM that generates the survey content and a helper LLM that critiques the generated surveys. The framework allows for incorporating human feedback as well, which can be useful when user-specific requirements need to be provided.

**Strengths:**

- The paper is clearly written, well-motivated, and easy to follow.
- The design of the FIKSurvey framework is intuitive, as it focuses on all important aspects of a survey (ie, structure, content, and factual grounding) through specific feedback modules.
- Quantitative results show clear improvements over existing SOTA methods like AutoSurvey and SurveyX in both content and citation quality.

**Weaknesses:**

- The paper is lacking standard deviations around scores in main result tables, I think this is important to establish the significance of the results. There is very little difference between ablation numbers (for eg, the content score differences in the ablation studies are small and likely not statistically significant). This has me wondering about the individual contribution of each module.
- There is no human evaluation of generated surveys conducted and the paper relies exclusively on an LLM-as-evaluator for all content quality metrics. I understand this may be hard, given that surveys are long, but an attempt at this is necessary for such a paper.
- LLM judge reliability is not established by comparison with human judges. This is a significant weakness, as the evaluator LLM may be systematically biased towards the style of the writer LLM. Human-rated surveys often score lower than generated surveys, which makes me further question the validity of LLM judge reliability.
- The paper is not exceptionally novel; at the core of it, the proposed framework uses a generator LM and a critique LM and iteratively improves survey quality.
- The models used (GPT-4o, GPT-4o-mini) are not open-source and may not continue to be served, which harms reproducibility of these results.
- Finally, the paper doesn't compare the latency of their framework (compute time, tokens generated) compared to baselines.

**Questions:**

- Could you report the survey length distributions for all systems? Are these substantially different and could LLM judges be biased due to survey length?
- Could you justify the hyperparameter choices made for Citation Feedback Correction (like the retrieval budget, window size) and how sensitive the results are to these parameters?
- Is the "Sliding-Window NLI" mechanism a novel contribution for ensuring factual grounding in retrieval-augmented generation?

---

### Note · Authors · 2025-12-03

I have read and agree with the venue's withdrawal policy on behalf of myself and my co-authors.